# Increasing Socially Significant Behaviors for Children with Autism Using Synchronous Reinforcement

Stephany K. Stordahl [1], Joseph H. Cihon [2],*, Shahla Alai-Rosales [3],* and Jesus Rosales-Ruiz [3]

1   Caravel Autism Health, 3936 Circle Drive, Holmen, WI 54636, USA; sstordahl@caravelautism.com
2   Autism Partnership Foundation, Seal Beach, CA 90740, USA
3   College of Health and Public Service, University of North Texas, Denton, TX 76205, USA;
    jesus.rosales-ruiz@unt.edu
*   Correspondence: jcihon@apfmail.org (J.H.C.); shahla.alai@unt.edu (S.A.-R.)

**Abstract:** In some types of reinforcement schedules, a reinforcer is presented or given to the child and interacted with continuously while a target behavior is occurring. Previous researchers have used this type of reinforcement as an assessment tool and to study basic reinforcement processes. However, little research has explored how to effectively use these schedules to reinforce socially important responses in applied settings. The current study analyzed the implementation of synchronous reinforcement in a nonconcurrent multiple baseline across participants. Several interventionists implemented synchronous reinforcement with four children with autism across a variety of responses and reinforcers. The results indicated that delivering social, audio/visual, and tangible stimuli on a synchronous schedule resulted in increased durations of targeted (e.g., social skills and motor skills) and nontargeted (e.g., approach, social bids, and speed) measures across all children. Recommendations regarding reinforcer and response selection in implementing synchronous schedules in applied settings are provided.

**Keywords:** autism; synchronous reinforcement; schedules of reinforcement





## 1. Increasing Socially Significant Behaviors for Children with Autism Using Synchronous Reinforcement

How to best employ reinforcement for the well-being of children with autism is, and should be, an important aim of intervention research. Reinforcement procedures involve the systematic use of consequences to increase desired responses and is considered one of the primary mechanisms of learning. Timing in relationship to the child's desired behavior is an essential component of effective reinforcement procedures. There are two basic ways to present reinforcement, after and during the child's responses.

The majority of treatment procedures employ episodic reinforcement, in which a reinforcer is presented and interacted with *after* a child responds. The child must stop responding and emit additional behavior (e.g., walking, reaching, eating, and watching a screen) in order to interact with the reinforcer. An alternative type of reinforcement, which is seldom used during treatment, involves a continuous, covarying response–reinforcer relation. This type of reinforcement schedule was coined "conjugate reinforcement" by Ogden Lindsley [1]. In a conjugate schedule of reinforcement, the reinforcer is coincidental with the occurrence of the response. That is, the reinforcer is interacted with *while* the individual responds, and no additional or competing response is required to interact with the reinforcer.

Conjugate reinforcement has been very useful to study the effects of sleep deprivation and sedatives on depth of sleep [2]. It has also been used to behaviorally monitor the effects of surgical anesthesia [3], duration and depth of electroshock therapy (EST)-induced comas [4], and the recovery from those states. Covarying reinforcement has also been used to assess the effects of different audio and video stimuli on participants' viewing

and listening behavior [1,5–8]. More recently, MacAleese and colleagues [9] assessed the effects of covarying reinforcement (clarity of pictures presented on a computer screen) on the key pressing of college students. Further, covarying reinforcement has been useful for studying social behavior and reinforcers [10–12], infants' learning and memory [13–17], and the analysis of stereotypic or self-stimulatory behavior [18–20].

Recently, Diaz de Villegas et al. [21] differentiated between two types of response-reinforcer covarying schedules: conjugate and synchronous. They said:

> ... in synchronous reinforcement, the behavior and reinforcer relation is all or nothing—if the behavior is happening, the reinforcer is delivered, whereas, if the behavior is not happening then the reinforcer is not delivered. In contrast, in conjugate reinforcement, some dimension of the behavior controls some dimension of the reinforcer—if the behavior is happening at a certain rate or intensity, then the reinforcer is delivered at that rate or intensity (p. 1661).

By this definition, conjugate schedules vary both the rate and the intensity of the reinforcer; for example, if a child is learning to write letters, music would play as long as they are practicing writing the letters and the music would increase in volume as they pressed with the desired amount of pressure as they wrote (rather than making barely visible marks). Synchronous schedules would involve only playing the music as the child made the letters. In a study with preschoolers with no known diagnoses, Diaz de Villegas et al. [21] investigated whether a synchronous schedule (covarying rate but not intensity) would be more or less effective than an episodic schedule (giving the reinforcer after the response) and whether participants would demonstrate a preference between these schedules. Participants were given access to high-preference music playlists either *during* (synchronous) or *after* (episodic) specified durations of on-task worksheet engagement. Each condition was associated with a color, and participants were provided with a choice between colors associated with each type of reinforcement (receiving after or during the activity). Participants showed a higher duration of task engagement and more preference for the synchronous (during) rather than the episodic (after) reinforcer arrangement.

Diaz de Villegas et al. [21] was replicated and extend by Hardesty et al. [22] by comparing synchronous and noncontingent schedules of reinforcement. They also used preferred music but compared music access during on-task behavior (synchronous) to just having the music available on a regular time interval and not dependent on the on-task behavior. In this case, also, the findings suggest the synchronous schedule more effectively increased on-task behavior in children who had no known diagnoses.

Both Diaz de Villegas et al. [21] and Hardesty et al. [22] demonstrated the utility of synchronous reinforcement with on-task behavior for children with no known diagnoses. As of now, episodic reinforcement is most frequently used to teach children with autism. In this system, the child engages in a response, and then a consequence is delivered. For example, in discrete trial teaching [23] the therapist delivers the instruction, "Time to do puzzles". After the child completes the puzzle, the therapist gives the child a preferred item (e.g., a book) or allows the child to engage in an activity (e.g., bubble blowing and popping). The child is given access to the item, the child interacts with the item, the item is removed, and the instruction is delivered again. In another common format, naturalistic instruction [24], high-preference items are identified and used as consequences after desired responding related to the high-preference item, activity, or event. For example, after the child requests a toy, the therapist will present the toy to the child. In general, many teaching methods in behavioral interventions operate under contingencies that involve the child engaging in a response and then receiving the reinforcer. While episodic reinforcement has been successful for teaching many behaviors with a variety of reinforcers, clinicians and educators often face difficulties with finding and arranging reinforcement for children with restricted activities and interests, such as autism [25]. Furthermore, when interests are restricted, it becomes difficult to find effective and lasting reinforcers.

The findings from Diaz de Villegas et al. [21] and Hardesty et al. [22] suggest that synchronous reinforcement holds promise for increasing our ability to understand the

parameters and effectiveness of reinforcement to support children diagnosed with autism. The purpose of the current study is to provide a demonstration of the effects of synchronous schedules of reinforcement on the duration of goal responses for four children diagnosed with autism. The reinforcers extended beyond music and included toys, electronic devices, stories, and social attention delivered by several therapists. Target goals in the synchronous contingency included the on-task behavior of sitting during lunch, but also extended to proximity to the therapist, proximity to peers, and stationary biking. Additionally, two goals were measured that were under no programmed contingency but are important to a child's development: social approaches and social bids.

## 2. Method

### 2.1. Participants and Setting

Four children diagnosed with autism spectrum disorder (ASD) were selected to participate in this study. All children received services at a nonprofit autism treatment center and were of varying skill levels. Each child was selected based on behaviors currently targeted in his or her individual dynamic programming. Table 1 provides an overview of each child's pertinent participant information, target goal responses, and specific consequences employed.

**Table 1.** Participant Information.

| Child | Age and Ethnicity | Diagnosis | Setting | Target Goal Responses | Non-Targeted Measures | Reinforcer |
|---|---|---|---|---|---|---|
| Max | 5.8 years old, Ethiopian | Autism | Kitchen | Duration sitting at lunch table | Frequency of approaches to table | Backyardigans[TM] book (Simon & Schuster, New York, NY, USA), Thomas the Train[TM] book (Random House, New York, NY, USA), V-tech[TM] toys (V-Tech, Hong Kong, China) |
| Emily | 7.8 years old, Asian Caucasian | Autism, Down syndrome, epilepsy | 1:1 therapy room | Proximity to therapist | Frequency of approaches to therapist | Leap Frog[TM] phonics radio, electronic shape computer toy (V-Tech, China) |
| Ulysses | 5.5 years old, Middle Eastern | Autism, mixed receptive language disorder | School room | Proximity to peers | Child social bids; peer social bids | iPhone© videos, e.g., Tom & Jerry[TM], Mickey Mouse[TM] Clubhouse, and Teletubbies[TM] (Apple, Cupertino, CA, USA) |
| Carl | 5.7 years old, Caucasian | Presumptive PDD-NOS; chromosomal deletion | Physical therapy gym | Bike pedaling | Speed (revolutions per minute) | Ghost stories told by therapist |

Sessions were conducted at various locations in the autism treatment center. Max's sessions took place in the kitchen, which included two tables, chairs, kitchen appliances (e.g., refrigerator), and a shelf with various toys. Emily's sessions occurred in a treatment room that contained a table, chairs, a one-way window that adjoined an observation room, and toys selected for the purpose of the study. A circle 1 m in diameter was marked with gray tape in the corner of the therapy room. Carl's sessions took place in the physical therapy (PT) gym, which contained various exercise equipment, bicycles, and toys secured in cabinets. One small bicycle attached to a CycleOps[TM] bike trainer was used for the purpose of the study. Ulysses' sessions occurred in the school room, which was primarily used for group and individual academic instruction. The school room included other children, tables and chairs, a circle time area, various toys on shelves, a bookshelf, and program materials for each child receiving services at the center.

### 2.2. Observation

The experimenter and/or the second observer recorded all sessions on a Flip Video© camera, and data were collected from the recorded videos. The experimenter measured

minutes engaged in the target behavior and nontargeted responses during all sessions for each participant, and a second observer independently scored 48% of total sessions for target behaviors and nontargeted responses. Responses included proximity, approach, social bids, and pedaling (defined below). The experimenter and observer independently scored target responses by recording the start and stop times for each onset and offset of the response (described below). Agreements and disagreements were calculated for each start and stop time. An agreement was recorded when the experimenter and observer recorded a start or stop time within 1 s of each other, while a disagreement was recorded when the experimenter and observer recorded a start or stop time greater than 1 s of each other. Interobserver agreement (IOA) was calculated by adding the total number of agreements, dividing by the agreements plus disagreements, and multiplying by 100. IOA for target responses across each participant ranged from 80% to 100%. Mean agreement for responses overall ranged from 91% to 100%.

Non-targeted responses were recorded separately. The experimenter and second observer recorded the frequency of Emily's approaches to the therapist and Max's approaches to the table within each session. Similarly, for Carl, the total number of bike wheel revolutions were recorded for each session. Agreement was indicated when the experimenter and observer recorded the same frequency of approaches or revolutions within the session. For Ulysses, child and peer social bids were scored using 10 s whole interval recording. An agreement was recorded when the experimenter and observer scored the same number of social bids in the interval. IOA for the nontargeted responses was calculated the same as the targeted responses. Mean IOA for nontargeted responses was 97% (range: 88% to 100%).

### 2.3. Procedures

#### 2.3.1. Experimental Design

A nonconcurrent multiple baseline across participants design [26] was used to examine the effects of a synchronous schedule of reinforcement on the duration each child engaged in the target response. Target responses were selected and individualized according to each child (i.e., Emily's proximity to the therapist, Ulysses' proximity to peers, Carl's bike pedaling, Max's sitting at the snack table). Due to staff and child availability, baseline lengths were predetermined and were assigned to children as they became available for the study.

#### 2.3.2. Preference Assessment

Preference assessments were individualized for each child. Stimuli were selected based on staff familiarity with the child's reinforcer history in the therapeutic setting. It should be noted that all items requested and presented to the children were not primary or symbolic reinforcers. They were all toys, music, or stories. Preferences are listed in Table 1.

One child who could vocalize preferences (i.e., Carl) was asked what he wanted (e.g., "Do you want to tell ghost stories or watch Bubble Guppies?") prior to each session in the synchronous condition [27,28]. After the participant selected an item, the therapist presented access to the item for 15 s. After 15 s, preferences were reassessed. If the child selected the same item after the initial 15 s, the item was used in the synchronous condition.

Children with limited vocal verbal repertoires (i.e., Max, Emily, and Ulysses) were presented with a multiple stimulus without replacement (MSWO) preference assessment [29]. An MSWO was implemented immediately before each session in the synchronous condition in a treatment room with various stimuli that were likely to function as reinforcers based on therapist anecdotal reports (e.g., books, V-tech$^{TM}$ Smart Writer, electronic toys).

#### 2.3.3. Baseline

Each baseline session lasted 15 min or ended when the child engaged in an escape response from the task or location (e.g., saying "all done"). No programmed consequences (e.g., praise, preferred item identified as identified in MSWO) were delivered contingent

on target responses during baseline. Thus, the therapist did not give the child their most preferred item at any time during the session.

### 2.3.4. Synchronous Condition

Each synchronous session lasted 15 min or concluded when the child engaged in an escape response from the task or location. In the synchronous session, the child was given continuous access to the most preferred item or activity identified in the preference assessment as long as the child engaged in the target response. Access to the item or activity was immediately discontinued when the participant stopped engaging in the target response. At the end of baseline and synchronous sessions, the therapist provided general praise (e.g., "Good job"), indicated to the child it was time to go on to the next activity, and then transitioned the child to the next scheduled event (e.g., circle time, game time).

### *2.4. Individual Procedures*
### 2.4.1. Max

Max was selected as a participant due to infrequent approaches to the table during snack time and not sitting at the table with his peers during lunch and snack time. Therefore, his target response included sitting in a chair at the table during snack time. Sitting was defined as Max's buttocks being in contact with the seat of the chair when the chair was positioned within 1 ft (0.3 m) of the table without engaging in challenging behavior (e.g., crying) identified in the child's individual programming. He could sit in any chair that was positioned within 1 ft (0.3 m) of the table. Approaches to the table were measured as a nontargeted response. An approach included Max walking directly toward the table or standing up from the chair and sitting down in any chair positioned next to the snack table. Each time Max walked toward or stood up from the chair and sat down in the chair was scored as one instance of approach. Max's preferred items used during synchronous sessions included Thomas the Train^TM books, Backyardigans^TM books, and V-Tech^TM toys.

The general session set-up remained consistent across baseline and synchronous sessions. Prior to each session, Max's most preferred items were removed from the kitchen, and his snack was placed on the table with a chair positioned in front of the snack table. Each session began with the therapist walking with Max to the kitchen from a 1:1 treatment room. When he arrived at the entrance to the kitchen, the therapist removed any items from his hands and placed them outside the kitchen door. The therapist established eye contact with Max and provided the instruction, "It's time for snack. Go sit down in the chair", and pointed toward the chair. After the therapist delivered the instruction, Max was able to move around the room (e.g., look out the window, turn the lights on and off). The therapist moved around the kitchen or stood next to the chair and observed Max but did not speak to him throughout the session. If he attempted to engage with kitchen appliances (e.g., press buttons on the microwave), engage in behavior that could be potentially dangerous (e.g., climbing on counter), or elope from the kitchen into the hallway or the adjacent room, the therapist stood in front of the location without making eye contact or delivering instructions (e.g., "Don't do that") and gently guided his shoulders away from the dangerous item or location.

During baseline sessions, the most preferred item was not available. If Max sat in any chair in the kitchen, the therapist remained standing next to the chair but did not provide social praise (e.g., "Good job") or interact with him.

During synchronous sessions, the therapist stood within 2 ft (0.6 m) of the chair that was positioned in front of Max's snack, with the exception of redirecting him away from areas in which he could engage with kitchen appliances (e.g., pressing buttons on the microwave), engage with potentially dangerous items, or attempt to leave. The therapist held his most preferred item and observed him throughout the session. The therapist handed Max the most preferred item (e.g., book, toy) and turned the item on, when applicable, immediately when he sat in any chair positioned next to the table. As long as Max was sitting, he had complete control of the item. While sitting in the chair, Max could set the item next to the chair, set the item in his lap, turn the item on or off, or manipulate

the item in any way. If Max stood up from the chair, the therapist removed the item from his hands and turned the item off (when applicable). If Max requested the item (e.g., looked at therapist and said "book") but was not sitting in the chair, the therapist said, "You need to sit down", and pointed toward the chair.

### 2.4.2. Emily

Emily was selected as a participant for the study because of infrequent social responding and approaches to the therapists. Her target response was proximity to the therapist, which was defined as the child sitting, standing, or lying in the designated circle with over half of her body crossing the plane of the circle without engaging in challenging behavior (e.g., pinching, head hitting). A non-example included lying on the floor with her head and trunk of body positioned outside of the circle. Approaches to the therapist were included as a nontargeted response. An approach was recorded when Emily entered the circle while the therapist was positioned inside. Each time Emily left the circle and entered the circle again was scored as one instance of approach. Emily's sessions took place in a 1:1 room. Emily's preferred items used in synchronous sessions included electronic toys that could be turned on and off such as musical keyboards, a Leap Frog™ radio, and various V-Tech™ toys.

The general set-up was similar throughout sessions in the baseline and synchronous conditions. A circle 3.5 ft (1 m) in diameter was marked with gray tape in the corner of the therapy room, which was used to ensure treatment fidelity across therapists. Prior to each session, low preferred stimuli (e.g., bead toy, puzzles, markers, books) were arranged around the room outside of the marked circle, and the most preferred items were removed and placed in an alternative treatment room. Stimuli that were not used for the purpose of the study were secured in cabinets. At the beginning of each session, the therapist led Emily to the 1:1 treatment room and delivered the initial instruction, "Come sit next to me". Emily could freely move around the room (e.g., walk around the room, look in the mirror) and engage with low-preference stimuli. The therapist sat in the back edge of the circle and observed Emily but did not say anything to her throughout the session.

During baseline sessions, Emily's most preferred electronic toy was not available at any time. The therapist remained seated in the back edge of the circle throughout the session. If Emily entered the marked circle, the therapist did not socially interact (e.g., provide eye contact, speak, give items) with her in any way.

Prior to each session in the synchronous session, an MSWO was conducted to determine the highest preferred item. During the synchronous sessions, the therapist remained in a fixed location, seated inside the edge of the circle holding Emily's most preferred electronic toy. The therapist turned the electronic toy (e.g., Leap Frog™ phonics radio) on and gave Emily the toy when she entered the marked circle. As long as Emily was sitting or standing in the marked circle, the toy remained on, and she could manipulate the toy in any way (e.g., press buttons, turn the toy on and off, set the toy down). If Emily stepped or crawled outside of the marked circle, the therapist removed the toy from her hands and turned the toy off. Throughout the session, Emily could move freely around the room and engage with low-preference items in the room, but the therapist only gave her the most preferred toy if she was positioned within the circle.

### 2.4.3. Ulysses

Ulysses was chosen as a participant in the study due to a low number of approaches to peers and infrequent social interactions with peers. His target response involved approaching a peer and sitting, crouching, or lying within 2 ft (0.6 m) of the peer, with over half of his body positioned within a 2 ft (0.6 m) radius of the peer without engaging in challenging behavior. A non-example included lying on the floor with his head positioned outside of a 2 ft (0.6 m) radius. Non-targeted responses included Ulysses' social bids and peer social bids. Ulysses' social bids included looking at the peer or engaging in a vocal request to obtain an item or access an activity (e.g., "I want the phone"), orienting head and eyes toward the peer's eyes, initiating or responding to joint attention with the peer, or attending

to the peer (e.g., looking at what peer is engaged with). Peer social bids were defined as the peer looking at the participant or engaging in a vocal request to obtain an item or access an activity (e.g., "Let's watch a different one"), orienting head and eyes toward Ulysses' eyes, initiating or responding to joint attention with Ulysses, initiating comments regarding the item the Ulysses was engaged with (e.g., "That's cool"), or attending to Ulysses (e.g., looking at which child). Ulysses' preferred items included physical play (e.g., tickles, spinning), watching videos on an iPhone©, sandboxes, and swinging.

Session set-up remained consistent during baseline and synchronous sessions. All items in the school room were left in their respective locations. The therapist approached Ulysses in the school room, obtained eye contact, pointed toward a peer, and said, "Go sit next to your friend". Ulysses was free to move around the school room (e.g., walk around the room, look out the window) and engage with any toys. The therapist moved around the school room and observed Ulysses but did not say anything or interact with him. If Ulysses attempted to engage in dangerous behavior (e.g., climbing on the counter) or attempt to leave the school room, the therapist redirected him away from the area by standing in front of the location without making eye contact or delivering instructions.

During baseline sessions, the therapist removed the Ulysses' most preferred item (e.g., iPhone©) from the school room. The therapist moved around the school room and observed Ulysses. He was allowed to interact with any items or people in the school room, but the therapist did not interact with him at any time during the session. If he approached a peer or remained in close proximity to a peer, the therapist did not interact him or the peer.

During the synchronous condition, one peer was selected for the purpose of the study. Throughout the session, the peer's therapist prompted him to remain in one area of the school room and engage with items in that area. The peer's therapist also prompted initiations toward Ulysses when appropriate (e.g., if the peer asked the therapist that Ulysses turn on a specific video on the iPhone) and delivered praise contingent on initiations toward Ulysses. Throughout sessions, Ulysses' therapist held his most preferred item (i.e., iPhone). After the initial instruction was delivered, the therapist moved around the school room and observed Ulysses but did not say anything or interact with him. When Ulysses moved within 2 ft (0.6 m) of the peer, the therapist turned the iPhone on and handed it to him. As long as Ulysses remained within 2 ft (0.6 m) of the peer, he could manipulate the iPhone in any way (e.g., select various videos, set the item down, turn the volume up), during which the therapist sat directly behind him and did not interact with him. If Ulysses left the 2 ft (0.6 m) radius in which the peer was positioned, the therapist turned the iPhone© off and removed the item from Ulysses' hands. Ulysses was free to move around the school room and engage with other items in the school room throughout each session, but he could only gain access to the iPhone© if positioned within 2 ft (0.6 m) of the peer. If Ulysses requested the iPhone© but was not engaged in the target response, the therapist delivered the initial instruction again.

### 2.4.4. Carl

Carl was chosen to participate in the study due to gross motor deficits and his family's desire to have him ride a bicycle. His target response was bike pedaling, which was defined as both of his feet making contact with both bike pedals, moving in a clockwise direction. A non-example included his feet making contact with both bike pedals and failing to move in a clockwise direction for longer than 1 s. Speed (in revolutions per min) was calculated during each session as a nontargeted measure by dividing the total number of revolutions by the total duration biked. One revolution was defined as Carl's feet making contact with both bike pedals, moving clockwise, and completing one 360 degree rotation from the starting point. Carl's sessions took place in a physical therapy (PT) gym. He selected ghost stories as his most preferred item prior to each synchronous session.

The initial set-up was consistent across all experimental sessions. All items in the PT gym were left in their typical locations. Prior to each session, the therapist told Carl that he was going to practice bike riding and transitioned with him to the PT gym. Once Carl sat

on the seat of the bike, the therapist said, "It's time to bike ride. Bike as long as you can as fast as you can". Carl was given the opportunity to bike throughout the 15 min session. The session ended after 15 min or when he engaged in an appropriate break response (e.g., saying, "I'm all done now").

During baseline sessions, the therapist stood within approximately 5 ft (1.5 m) of the bike and observed Carl ride the bicycle, but did not speak to him at any time during the session. If Carl attempted to speak to the therapist, the therapist did not respond. The therapist did not tell ghost stories at any time during the session regardless of whether Carl was or was not biking.

During sessions in the synchronous session, the therapist stood in front of the bicycle Carl was sitting on. Throughout the session, the therapist observed Carl's feet on the bike pedals. When he began biking, the therapist looked at him, displayed positive affect, and began telling ghost stories. Ghost stories involved Carl's selection of characters (e.g., Mr. Joe) and setting (e.g., the woods). The ghost stories involved a series of suspenseful incidents, sound effects, and various facial expressions. The therapist continued to tell ghost stories as long as Carl was biking. If Carl stopped biking, the therapist looked down at the bike pedals, displayed neutral affect, and stopped telling the ghost story. If Carl began biking again, the therapist looked at him, displayed positive affect, and began telling the story from where the story was left off. Carl was free to stop at any time and engage with other items in the PT gym, but the therapist only told ghost stories and looked at him when he was engaging in the target response. If Carl requested ghost stories when he was not biking, the therapist gestured toward the bike and said, "It's time to ride the bike".

### 3. Results

Figure 1 displays the results for the duration of the target response and the frequency of nontargeted responses (e.g., approach, social bids toward peers, and pedaling speed) across experimental sessions for each participant. During the baseline condition, each participant's engagement in the target response remained low, with Max engaging in the target response for less than 8 s, Emily less than 1 min, Ulysses less than 15 s, and Carl less than 20 s. When the synchronous condition was introduced, engagement in the target response increased across all participants. Specifically, Max's engagement in the target response ranged from approximately 6.5 min to 14.5 min. Emily exhibited a sustained duration of engagement in the targeted response at nearly 15 min across sessions in the synchronous condition. Ulysses' duration of engagement in the targeted response remained above 12 min, and Carl's target response increased up to 3.5 min over baseline levels.

Implementation of the synchronous condition resulted in changes in additional, non-targeted responses across participants. During the baseline condition, Max approached the table one time. When the synchronous condition was introduced, the frequency of Max's approach to the table increased to 11 in Sessions 2 and 4 and remained at 5 approaches or greater for the remainder of sessions. Emily engaged in four or fewer approaches to the therapist during baseline. The synchronous condition produced a variable number of approaches, initially increasing to eight approaches in Session 4 and ranging from four to nine approaches in Sessions 5–9. Frequency of social bids during Ulysses' baseline sessions remained at three or fewer, and peer social bids remained low. When the target response was synchronously reinforced, Ulysses' social bids initially increased to nine, eight, and nine in Sessions 6, 7, and 8, respectively, and decreased to two or fewer in Sessions 9 and 10. Peers social bids increased, ranging from 58 to 85 in the synchronous condition. Carl's bike pedaling speed was variable in baseline, with an increasing trend from Sessions 4 to 7. The speed of Carl's bike pedaling exhibited a decreasing trend in the synchronous condition.

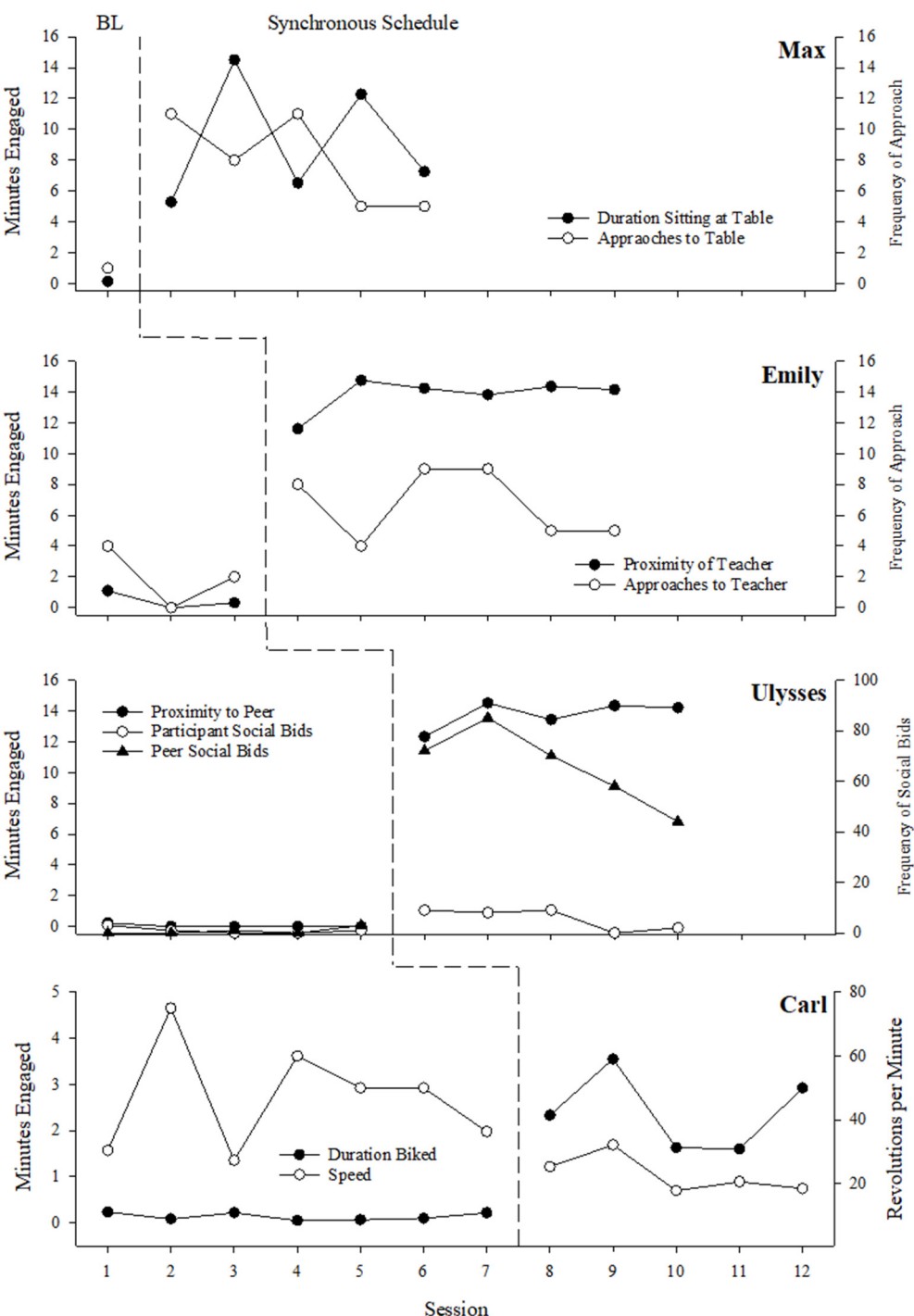

**Figure 1.** Duration of targeted goal responses and nontargeted measures across participants.

## 4. Discussion

The results show that delivering reinforcers on a synchronous schedule produced increases in the duration of all target responses. The duration of Emily's sitting or standing next to the therapist and Ulysses' duration of sitting next to the peer increased when reinforced on a synchronous schedule. Likewise, Max sat at the lunch table for longer durations and Carl's duration of bike pedaling increased. Also, approaches to the lunch table for Max increased, and approaches to the peer and therapist increased for Ulysses. These data extend the generality of synchronous reinforcement to new reinforcers and behaviors [21,22] and show that synchronous reinforcement can be useful to increase and maintain behavior in applied practice with children diagnosed with autism.

In addition to increases in the duration of target behaviors, the implementation of the synchronous schedules brought about desirable behavior changes that were not directly targeted. These changes were likely a result of increasing proximity to a community of reinforcement as a result of the contingency (e.g., peers). Moreover, it has been shown that reinforcing responses resulting in proximity to a community of reinforcement could maintain behavior beyond the entry response and may further shape the individual's behavior in new directions [30–32]. However, in other cases, proximity was not enough to introduce children to communities of reinforcement. For example, in this study, Ulysses' social bids toward the peer were near zero prior to the synchronous condition. During the synchronous condition, social bids increased immediately and substantially but did not maintain, despite maintaining basically the same conditions that produced this change originally. Social bids began to decline after the third session of synchronous reinforcement, but did remain substantially higher than during baseline sessions. A number of factors may have contributed to the decrease in social bids. One, Ulysses chose the same videos during each session. Therefore, satiation may have played a role in the peer's responding due to lack of variability in videos. Two, the decline coincides with the introduction of less experienced, novel therapists. During the initial implementation, a more experienced therapist directed the peer's initiations toward Ulysses and provided a denser schedule of reinforcement for interactions.

The implementation of synchronous schedules requires careful consideration of the selection of the response, the reinforcer, and the delivery of the reinforcer. The reinforcer should be one that the therapist can easily manipulate. To ensure a synchronous schedule is implemented correctly, the therapist must turn the reinforcer "on" and "off" immediately as the child starts and stops engaging in the targeted response. Hence, electronic stimuli and social reinforcers are highly amenable to precise control of onset and offset of reinforcer delivery. At the same time, the switch toys and stories were easy to stop and start. The ghost stories were particularly compelling as both the therapist and the child seemed to enjoy the suspense of brief pauses before riding and storytelling resumed. For these reasons and to assist in developing a technology around synchronous reinforcement, future research should include measures of procedural integrity to determine the precision of implementation.

Since reinforcer interaction occurs at the same time as the response, interacting with the reinforcer should not prevent the child from engaging in the target response. A reinforcer that competes with the target response, such as using visual stimuli to increase attending to academic tasks, may be ineffective when implemented synchronously. One should also consider selecting reinforcers that can sustain performance and resist satiation for long durations. Auditory and visual stimuli, such as music and light toys, are good possibilities [6,7]. For example, Morgan and Lindsley [7] showed no satiation effects when covarying music for seven experimental hours.

With respect to the response selection, the continuous response–reinforcer relations required by synchronous contingencies require a response, or a collection of responses, that could be emitted continuously, without disruption by the reinforcer. Thus, any response that needs to be increased in duration or rate could be a candidate for synchronous reinforcement. Lindsley [33] suggested that covarying schedules of reinforcement are the most prominent schedule in the natural environment and that social responding and reinforcers were covarying in nature. When two individuals engage in a conversation, the rate of one person's talking may be a function of the rate at which the other person makes eye contact or displays certain facial expressions. For example, Lindsley [11] found that covarying social reinforcers (e.g., favorable facial expressions) were effective in increasing the rate of responding. Incorporating synchronous social reinforcers in autism intervention may be beneficial due to the social skills concerns and the lack of social stimuli functioning as reinforcers, which is characteristic in children diagnosed with autism. This notion is further supported by work on embedded social reinforcement [34]. The present study demonstrated effectiveness with both social and non-social responding and suggests that synchronous reinforcement has utility across response types. For children with a diagnosis

of autism, it is perhaps most important to explore its use for social behaviors. Furthermore, it may be that the units of responding could be reconsidered and explored. For example, discrete responses, such as answering a question, may employ episodic reinforcement but social responses such as having a reciprocal conversation might be better suited to synchronous social reinforcement.

The current study did not manipulate the magnitude of the reinforcer in relation to the performance. For example, the therapist told ghost stories to Carl while he was pedaling the bike, but the volume or speed of the ghost stories did not increase contingent on higher speeds of bike pedaling and decrease with lower speeds of bike pedaling. As a result, Carl's duration of biking increased, but his speed of biking decreased. Adjusting the reinforcer contingent on his speed may have resulted in higher speeds of bike pedaling. In a related study, Voltaire et al. [8] demonstrated that a higher incidence of peak responding occurred under conjugate reinforcement (e.g., yoked intensity) compared to responding reinforced regardless of intensity (e.g., continuous reinforcement). Future research with children with autism could also explore the effects of reinforcer magnitude. For instance, if the child had gained access to a more intense stimulus (e.g., louder volume) when seated within 1 ft (0.3 m) of the peer and a less intense stimulus (e.g., lower volume) when seated within 2 ft (0.6 m) of the peer, the child may have remained in closer proximity to the peer for longer durations. At the same time, consideration would need to be given to the content of the storytelling and if speeding up the story would retain the aspects that were enjoyable to Carl.

Additionally, it is important to note that this study took place in a clinical setting with children's actual intervention goals and with therapists that were familiar with the children. This is both an advantage and a disadvantage. On one hand, it is a demonstration of the effectiveness of the procedures in a clinical, not research, setting. On the other hand, there were probably more uncontrolled variables than in a research setting. For example, most reinforcement systems in educational and clinical settings are used in combination with prompting procedures (as was the case for Ulysses); this dimension is another aspect that was not controlled as it might have been in a basic research setting. Also, the observation periods were longer than in previous research [21,22]. This may extend the generality and strengthen the possibilities for applications in clinical and educational settings. Finally, in none of the children's cases was thinning the schedule of reinforcement or transferring reinforcer control explored. Clinically, each child had additional skills added to the teaching programs with the same schedules of reinforcement in place as described here.

Overall, the results of this study show that synchronous reinforcement can be an effective tool in an ongoing autism treatment program. Its effectiveness was replicated across children, responses, various reinforcing stimuli, and settings with several different interventionists implementing the procedures. Future research should explore under what conditions it may be more desirable or feasible to use synchronous reinforcement versus when episodic reinforcement or conjugate reinforcement may be more useful. This analysis would help clinicians and educators make better informed treatment decisions and design more effective reinforcement procedures, ultimately leading to improved client outcomes.

**Author Contributions:** Conceptualization, S.K.S., S.A.-R., J.H.C. and J.R.-R.; methodology, S.K.S., S.A.-R., J.H.C. and J.R.-R.; validation, S.K.S., S.A.-R., J.H.C. and J.R.-R.; formal analysis, S.K.S., S.A.-R. and J.R.-R.; investigation, S.K.S. and J.H.C.; data curation, S.K.S., S.A.-R., J.H.C. and J.R.-R.; writing—original draft preparation, S.K.S.; writing—review and editing, S.A.-R., J.H.C. and J.R.-R.; supervision, S.A.-R., J.H.C. and J.R.-R.; project administration, S.A.-R. All authors have read and agreed to the published version of the manuscript.

**Funding:** This research received no external funding.

**Institutional Review Board Statement:** The current study has been approved by the institutional review board committee and has been performed in accordance with the ethical standards laid down in the 1964 Declaration of Helsinki and its later amendments or comparable ethical standards.

**Informed Consent Statement:** Informed consent was obtained from all subjects involved in the study.

**Data Availability Statement:** The data presented in this study are available on request from the corresponding authors.

**Conflicts of Interest:** The authors declared no conflict of interest with respect to the authorship or the publication of this article.

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
