# Peer review of "Increasing Socially Significant Behaviors for Children with Autism Using Synchronous Reinforcement"

_education, doi:10.3390/educsci13070751_

Round 1

Reviewer 1 Report

Dear author, first of all, I would like to congratulate you on your dedication to your work. You might consider including an explanation of reinforcement schedules in the introduction. I suggest you provide clearer information about the main subject of your study, especially synchronous reinforcement. I suggest you add a few more sentences to explain your method more clearly. In addition, the types of reinforcers used in the subjects can be added to the relevant table (primary reinforcer, symbol reinforcer, etc.).

Dear author, first of all, I would like to congratulate you on your efforts. I recommend that you review your work in terms of concepts. Please consider rewriting concepts such as "deliver", "consume" "engineer" etc. with concepts suitable for educational sciences.

Reviewer 2 Report

See attached suggestions document. 

Author Response

Thank you so much for your careful review and suggestions. We revised as best as we could and noted the revisions in response to each of the reviewers’ comments. We were especially grateful for those suggestions that directed us to use language more understandable and useful.

Reviewer 1

You might consider including an explanation of reinforcement schedules in the introduction.

Added to the first paragraph: Reinforcement procedures involve the systematic use of consequences to increase desired responses and is considered one of the primary mechanisms of learning.

I suggest you provide clearer information about the main subject of your study, especially synchronous reinforcement.

Added the sentence above and also added: Timing in relationship to the child’s desired behavior is an essential component of effective reinforcement procedures. There are two basic ways to present reinforcement, after and during the child’s responses.

I suggest you add a few more sentences to explain your method more clearly.

Added more detail to Diaz de Villegas et al. (2020) and Hardesty studies in the introduction and added specific details to methods as suggested by reviewer 1.

In addition, the types of reinforcers used in the subjects can be added to the relevant table (primary reinforcer, symbol reinforcer, etc.).

Added: It should be noted that all items requested and presented to the children were not primary or symbolic reinforcers. They were all toys, music, or stories. Preferences are listed in Table 1.

I recommend that you review your work in terms of concepts. Please consider rewriting concepts such as "deliver", "consume" "engineer" etc. with concepts suitable for educational sciences.

Changed throughout to

  • delivered to “presented or given to the child”
  • consumed to “interacted with”
  • engineered to “use” or “designed”

Reviewer 2

Building a stronger rationale for the current study and highlighting it’s novel contributions

We added more description to the introduction and rearranged the sequence of the last paragraph in the introduction and the last paragraph of the discussion.

Reorganizing the study into two parts to focus on application to non-social and then social behaviors separately, therein highlighting the importance of the latter

It is an important point that there were both motor and the social responses. Changing the order of the graph would also change the staggering of the multiple baseline. Also, the goals were selected based on specific child goals. To address this issue however, we revised the following paragraph in the discussion: “The present study demonstrated effectiveness with both social and non-social responding and suggests that synchronous reinforcement has utility across response types. For children with a diagnosis of autism, it is perhaps most important to explore its use for social behaviors.”

Collect or provide procedural integrity data to characterize the degree to which the interventions in the current study truly constitutes synchronous reinforcement.

Given the revision timeline we are not able to go back and rescore the footage. We added this as a limitation to the discussion: “For these reasons and to assist in developing a technology around synchronous reinforcement, future research should include measures of procedural integrity to determine the precision of implementation”

Page 2, line 49: It seems their might be a [For] or something of the like missing from the excerpt of the Diaz de Villegas article.

Added “in”

Page 2, line 55: “worked with” seems a bit strange, I would suggest the authors consider an alternative. Similarly, it may be better to exchange “typically developing” for something like preschoolers with no known diagnoses.

Changed to: “In a study with preschoolers with no know diagnoses, Diaz de Villegas et al. (2020) investigated whether a synchronous schedule…

Page 2, line 61: the sentence “were also asked to choose their preferences for reinforcer delivery” could benefit form revision. Perhaps, “participants were provided with a choice between the colors associated with each type of reinforcer delivery”.

Changed to: participants were provided with a choice between colors associated with each type of reinforcement (receiving after or during the activity).

The Hardesty et al., (2023) study seems like it could benefit from a bit of additional information. It’s reviewed so quickly now that it’s a bit confusing.

Changed to: Diaz de Villegas et al. (2020) was replicated and extend by Hardesty et al (2023) by comparing synchronous and non-contingent schedules of reinforcement. They also used using preferred music as consequences but compared music access during on task-behavior (synchronous to just having the music available periodically and not dependent on the children’s on task behavior. In this case, also, the findings suggest the synchronous schedule more effectively increased on-task behavior.

Page 2, line 68: “one population and one set of behaviors” could benefit from more detailed/precise language.

Changed to: Diaz de Villegas et al. (2020) and Hardesty et al (2023) demonstrated the utility of synchronous reinforcement with on task behavior for children with no known diagnoses. It is unclear if this is effective for use with children with an autism diagnosis and if it will be effective for behaviors other than on task engagement and with music as the reinforcer. As of now, episodic reinforcement

Page 2, line 73: describing the initial instruction as a discriminative stimulus rather than a cue would seem more conceptually systematic, at least to me.

Changed to “instruction”.

The mention of the premack principle seems to kind of come out of nowhere. I’m not sure if it is necessary, but if the authors decide to include it, then it may be helpful to gowith the more modern, comprehensive response deprivation hypothesis (Timberlake & Allison, 1974).

We removed the Premack as it is not directly relevant.

Additionally, rather than the somewhat obscure examples provided, it could be helpful to cite some recent examples (or even reviews) of common interventions strategies for ubiquitous behavioral goals that incorporate episodic reinforcement (e.g., FCT, imitation training).

We reduced the episodic examples to two reviews (Smith 2001 as this was an example of DTI and Schreibman et al 2015 as this was an example of NDBI) that are the most common forms of intervention.

It seems the rationale for the current study could be significantly stronger. Right now the only point of rationale that’s given is “nobody has done this with kids with autism”. Which is something but could still be fleshed out more. For example, given the diagnostic criteria of autism, why may synchronous reinforcement exert different effects than those observed with children with no diagnoses in the Diaz de Villegas and Hardesty studies. Why may it be particularly beneficial for children with autism?

Added: While episodic reinforcement has been successful for teaching many behaviors with a variety of reinforcers, clinicians and educators often face difficulties with finding and arranging reinforcement for children with restricted activities and interests, such as autism (Alai-Rosales, et al 2008). Furthermore, when interests are restricted, it becomes difficult to find effective and lasting reinforcers.

Related to my preceding point, I think the use of different types of reinforcers beyond music, the evaluation of indirect effect on non-target measures that were not directly targeted, and the focus on social target behaviors for two participants are all unique characteristics of this study that have not been evaluated in previous research on synchronous reinforcement, which should be made more apparent prior to the statement of purpose.

The findings from Diaz de Villegas et al. (2020), Hardesty et al (2023) suggest that synchronous reinforcement holds promise for increasing our ability to understand the parameters and effectiveness of reinforcement to support children diagnosed with autism. The purpose of the current study is to provide a demonstration of the effects of synchronous schedules of reinforcement on the duration of goal responses for four children diagnosed with autism. The reinforcers extended beyond music and included toys, electronic devices, stories, and social attention delivered by several therapists. Target goals in the synchronous contingency included the on-task behavior of sitting during lunch, but also extended to proximity to the therapist, proximity to peer, and stationary biking. Additionally, two goals that were under no programmed contingency, but are important to a child’s development were measured: social approaches and social bids.

Method:

The authors should consider reorganizing the manuscript into two parts. The first could focus on the application of synchronous reinforcement with non-social target behaviors (Max and Carl). The novel contributions of the first part would lie in the use of reinforcers other than music in synchronous schedules of reinforcement. The second part could focus on the application of synchronous reinforcement to social target behaviors (Emily and Ulysses), and the authors could go into a bit more detail about the relevance of social target behaviors for children with ASD and the applicability of synchronous reinforcement for social behavior.

It is an important point that there were both motor and the social responses. Changing the order of the procedures and graph would also change the staggering of the multiple baseline. Also, the goals were selected based on specific child goals. To address this issue however, we revised the following paragraph in the discussion: “The present study demonstrated effectiveness with both social and non-social responding and suggests that synchronous reinforcement has utility across response types. For children with a diagnosis of autism, it is perhaps most important to explore its use for social behaviors.”

At least to me, it seems like it may be clearer to provide response definitions in the observation section of the manuscript.

Not sure what to change. The general response was included in Table 1 and the response definitions are in the observation section.

If a particular questionnaire (e.g., RAISD, Fisher et al., 1996) or other method was used for staff nomination of potential reinforcers, then it may be worth specifying this information.

The staff were working with the children and had ongoing preferences assessment and were tested prior to working with the child. References added and the wording was changed to indicate this:

Preference assessments were individualized for each child. Stimuli were selected based on staff familiarity with the child’s reinforcer history in the therapeutic setting. It should be noted that all items requested and presented to the children were not primary or symbolic reinforcers. They were all toys, music, or stories. Preferences are listed in Table 1.

Children who could vocalize preferences (i.e., Carl) were asked what they wanted (e.g., “Do you want to tell ghost stories or watch Bubble Guppies?”) prior to each session in the synchronous condition (Fisher et al., 1992; Tessing et al., 2006). After the participant selected an item, the therapist presented access to the item for 15 s. After 15 s, preferences were reassessed. If the child selected the same item after the initial 15 s, the item was used in the synchronous condition.

It may be worthwhile to provide a citation for the vocal PSPA, given there have been a few studies that demonstrated the utility of this approach for identifying reinforcers (Northup et al., 1996; Tessing et al., 2006; Morris & Vollmer, 2020).

Revised: Preference Assessment

Preference assessments were individualized for each child. Stimuli were selected based on staff familiarity with the child’s reinforcer history in the therapeutic setting. It should be noted that all items requested and presented to the children were not primary or symbolic reinforcers. They were all toys, music, or stories. Preferences are listed in Table 1.

Children who could vocalize preferences (i.e., Carl) were asked what they wanted (e.g., “Do you want to tell ghost stories or watch Bubble Guppies?”) prior to each session in the synchronous condition (Fisher et al., 1992; Tessing et al., 2006). After the participant selected an item, the therapist presented access to the item for 15 s. After 15 s, preferences were reassessed. If the child selected the same item after the initial 15 s, the item was used in the synchronous condition.

Added reference for paired stimulus preference assessment (Fisher, W., Piazza, C. C., Bowman, L. G., Hagopian, L. P., Owens, J. C., & Slevin, I. (1992). A comparison of two approaches for identifying reinforcers for persons with severe and profound disabilities. Journal of Applied Behavior Analysis, 25(2), 491–498).

Added reference for vocal preference assessment (Tessing, J. L., Napolitano, D. A., McAdam, D. B., Dicesare, A., & Axelrod, S. (2006). The effects of providing access to stimuli following choice making during vocal preference assessments. Journal of Applied Behavior Analysis, 39(4), 501 - 506. https://doi.org/10.1901/jaba.2006.56-05)

Unfortunately, the description of the measures for each participant make it clear that the non-target responses are inextricably related to the occurrence of the target response for everyone but Ulysses. Therefore, the authors may consider downplaying their significance.

Added a few changes to first paragraph on page 9 to contextualize: In addition to increases in the duration of target behaviors, the implementation of the synchronous schedules brought about desirable behavior changes that were not directly targeted. These changes were likely a result of increasing proximity to a community of reinforcement as a result of the contingency (e.g., peers). Moreover, it has been shown that reinforcing responses resulting in proximity to a community of reinforcement could maintain behavior beyond the entry response and may further shape the individual’s behavior in new directions (Baer & Wolf, 1970; Baer, Rowbury, & Goetz, 1976, Rosales-Ruiz & Baer, 1997). However, in other cases proximity is not enough to introduce children to communities of reinforcement. For example, in this study Ulysses’ social bids toward the peer were near zero prior to the synchronous condition. During the synchronous condition, social bids increased immediately and substantially but did not maintain, despite maintaining basically the same conditions that produced this change originally. Social bids began to decline after the third session of synchronous reinforcement, but did remain substantially higher than during baseline sessions. A number of factors may have contributed to the decrease in social bids. One, Ulysses chose the same videos during each session. Therefore, satiation may have played a role in the peer’s responding due to lack of variability in videos. Two, the decline coincides with the introduction of less experienced, novel therapists. During the initial implementation, a more experienced therapist directed the peer’s initiations toward Ulysses and provided a denser schedule of reinforcement for interactions.

In the description of the procedures for Ulysses, the statement, “Throughout sessions during the synchronous condition, Ulysses’ therapist held his most preferred item (i.e., iPhone)” is a bit confusing given that the therapist was not holding the item whenever it was delivered contingent on peer proximity. This sentence should be revised, and the authors should check for similar issues of clarity in description of the other participants’ procedures.

Changed to: Throughout sessions, Ulysses’ therapist held his most preferred item (i.e., iPhone).

Given the additional prompts and potential reinforcers being implemented for the peer initiations, it may be best to remove those data from Ulysses evaluation, given that they don’t reflect the influence of the synchronous reinforcement contingency.

We left this is because most reinforcement procedures in clinical practice will also involve prompts. At the same time, it is an important variable and we addressed it in the discussion:

Additionally, it is important to note that this study took place in a clinical setting with children’s actual intervention goals and with therapists that were familiar with the children. This is both an advantage and disadvantage. On one hand, it is a demonstration of the effectiveness of the procedures in a clinical, not research, setting. On the other hand there were probably more uncontrolled variables than in a research setting. For example, most reinforcement systems in educational and clinical settings are used in combination with prompting procedures (as was the case for Ulysses) and that this dimension is another aspect that was not controlled as it might have been in a basic research setting.

The inclusion of 15 min sessions is also a unique characteristic of this study that merits discussion. In the Diaz de Villegas study, sessions were only five minutes, so the authors may discuss that their evaluation demonstrated that synchronous reinforcement can be effective when implemented continuously over larger blocks of time.

Added: Also, the observation periods were longer than previous research (Diaz de Villegas et al., 2020; Hardesty et al., 2023). This may actually extend the generality and strength to the use in clinical and educational settings

Previous research has used music as reinforcement in synchronous schedules because of the ease with which it can be added in and removed from the environment. The current study utilized different forms of reinforcement, which is novel and interesting, but also certainly much more difficult to implement with integrity.

Revised to: The implementation of synchronous schedules requires careful consideration of the selection of the response, the reinforcer, and the delivery of the reinforcer. The reinforcer should be one that the therapist can easily manipulate. To ensure a synchronous schedule is implemented correctly, the therapist must turn the reinforcer “on” and “off” immediately as the child starts and stops engaging in the targeted response. Hence, electronic stimuli and social reinforcers are highly amenable to precise control of onset and offset of reinforcer delivery. At the same time, the switch toys and stories were easy to stop and start. The ghost stories were particularly compelling as both the therapist and the child seemed to enjoy the suspense of brief pauses before riding and storytelling resumed. For these reasons and to assist in developing a technology around synchronous reinforcement, future research should include measures of procedural integrity to determine the precision of implementation.

One change I believe the authors have to make prior to publication is to collect and include procedural integrity data to demonstrate the extent to which they were successful in truly implementing synchronous schedules of reinforcement. Measures such as they latency from the occurrence of target behavior to the onset of reinforcement and the latency form the offset of target behavior to the removal of reinforcement could be really informative.

Given the revision timeline we are not able to go back and rescore the footage. We added this as a limitation to the discussion: “For these reasons and to assist in developing a technology around synchronous reinforcement, future research should include measures of procedural integrity to determine the precision of implementation”

The communities of reinforcement points are interesting, but a bit unclear to me. The authors may consider unpacking this further. Perhaps discussing these points in terms of establishing others as discriminative for reinforcement would be a clearer and more conceptually systematic way to make the point.

Revised: In addition to increases in the duration of target behaviors, the implementation of the synchronous schedules brought about desirable behavior changes that were not directly targeted. These changes were likely a result of increasing proximity to a community of reinforcement as a result of the contingency (e.g., peers). Moreover, it has been shown that reinforcing responses resulting in proximity to a community of reinforcement could maintain behavior beyond the entry response and may further shape the individual’s behavior in new directions (Baer & Wolf, 1970; Baer, Rowbury, & Goetz, 1976, Rosales-Ruiz & Baer, 1997). However, in other cases proximity is not enough to introduce children to communities of reinforcement. For example, in this study Ulysses’ social bids toward the peer were near zero prior to the synchronous condition. During the synchronous condition, social bids increased immediately and substantially but did not maintain, despite maintaining basically the same conditions that produced this change originally. Social bids began to decline after the third session of synchronous reinforcement, but did remain substantially higher than during baseline sessions. A number of factors may have contributed to the decrease in social bids. One, Ulysses chose the same videos during each session. Therefore, satiation may have played a role in the peer’s responding due to lack of variability in videos. Two, the decline coincides with the introduction of less experienced, novel therapists. During the initial implementation, a more experienced therapist directed the peer’s initiations toward Ulysses and provided a denser schedule of reinforcement for interactions.

I appreciate the authors’ discussion of response-reinforcer considerations. However, I think the response part of this equation may be a bit underrepresented in this discussion. Synchronous reinforcement seems particularly well suited for reinforcing behaviors related to proximity or remaining in a given area such as those evaluated in the current study. On the other hand, it may not be a good reinforcer arrangement for behaviors that involve moving away from or independently of the therapist (i.e., instruction following) or that have a fleeting duration (i.e., communication responses).

Revised: With respect to the response selection, the continuous response-reinforcer relations required by synchronous contingencies require a response, or a collection of responses, that could be emitted continuously, without disruption by the reinforcer. Thus, any response that needs to be increased in duration or rate could be a candidate for synchronous reinforcement. Lindsley (1956) suggested that covarying schedules of reinforcement are the most prominent schedule in the natural environment and that social responding and reinforcers were covarying in nature. When two individuals engage in a conversation, the rate of one person’s talking may be a function of the rate at which the other person makes eye contact or displays certain facial expressions. For example, Lindsley (1963) found that covarying social reinforcers (e.g., favorable facial expressions) were effective in increasing the rate of responding. Incorporating synchronous social reinforcers in autism intervention may be beneficial due to the social skills concerns and the lack of social stimuli functioning as reinforcers, which is characteristic in children diagnosed with autism. This notion is further supported by work on embedded social reinforcement (Koegel, Vernon, & Koegel, 2009). The present study demonstrated effectiveness with both social and non-social responding and suggests that synchronous reinforcement has utility across response types. For children with a diagnosis of autism, it is perhaps most important to explore its use for social behaviors. Furthermore, it may be that the units of responding could be reconsidered and explored. For example, discrete responses, such as answering a question may employ episodic reinforcement but social responses such as having a reciprocal conversation might be better suited to synchronous social reinforcement.

Is it possible to thin the schedule of synchronous reinforcement? Or would not delivering the reinforcer for the entirety of the behavior’s occurrence make it something other than synchronous reinforcement? Given the importance of schedule thinning for feasibility and maintenance, these points would seem to merit discussion.

Revised to: Finally, in none of the children’s cases was thinning the schedule of reinforcement or transferring reinforcer control explored. Clinically, each child had additional skills added to the teaching programs with the same schedules of reinforcement in place as described here.

The data for Emily and Ulysses reminded me of research on “assessments of sociability” (e.g., Morris & Vollmer, 2022) in which social interaction is continuously available for being on the same side of the room as the therapist. These studies seem to be the closest approximation to synchronous schedules of reinforcement implemented for social target behaviors (i.e., proximity/time allocation) and with children with ASD. Moreover, the study I cited above actually produced interesting results parallel to those of the current study: delivering more preferred, individualized social reinforcer contingent on social proximity increased social proximity, increased mands for social play, increased instances of social approach, and decreased instances of social avoidance. As such, I think it may be beneficial to discuss your findings in relation to this research, doing so could improve the impact of the current study

We are not sure how to incorporate this as the two studies we found by Morris and Vollmer 2022 both appeared to use episodic reinforcement. The procedure you described would be similar to synchronous but perhaps we were looking at the wrong studies. If possible, could you include the whole reference so that we can incorporate it? It seems like a really great procedure that would enhance, as you said, the importance of this study.
